# Inhibition of *Cxcr4* Disrupts Mouse Embryonic Palatal Mesenchymal Cell Migration and Induces Cleft Palate Occurrence

**DOI:** 10.3390/ijms241612740

**Published:** 2023-08-13

**Authors:** Xiaoyu Zheng, Xige Zhao, Yijia Wang, Jing Chen, Xiaotong Wang, Xia Peng, Li Ma, Juan Du

**Affiliations:** Laboratory of Orofacial Development, Laboratory of Molecular Signaling and Stem Cells Therapy, Molecular Laboratory for Gene Therapy and Tooth Regeneration, Beijing Key Laboratory of Tooth Regeneration and Function Reconstruction, Capital Medical University School of Stomatology, Tiantan Xili No. 4, Beijing 100050, China; 18519220100@163.com (X.Z.); 18842663996@163.com (X.Z.); yijiawang1994@163.com (Y.W.); chenjingecho629@163.com (J.C.); wangxiaotongkx98@163.com (X.W.); 18834183343@163.com (X.P.); 15699978899@163.com (L.M.)

**Keywords:** C-X-C motif chemokine receptor 4, palate development, cleft palate, migration

## Abstract

Many processes take place during embryogenesis, and the development of the palate mainly involves proliferation, migration, osteogenesis, and epithelial–mesenchymal transition. Abnormalities in any of these processes can be the cause of cleft palate (CP). There have been few reports on whether C-X-C motif chemokine receptor 4 (CXCR4), which is involved in embryonic development, participates in these processes. In our study, the knockdown of *Cxcr4* inhibited the migration of mouse embryonic palatal mesenchymal (MEPM) cells similarly to the use of its inhibitor plerixafor, and the inhibition of cell migration in the *Cxcr4* knockdown group was partially reversed by supplementation with C-X-C motif chemokine ligand 12 (CXCL12). In combination with low-dose retinoic acid (RA), plerixafor increased the incidence of cleft palates in mice by decreasing the expression of *Cxcr4* and its downstream migration-regulating gene Rac family small GTPase 1 (RAC1) mediating actin cytoskeleton to affect lamellipodia formation and focal complex assembly and ras homolog family member A (RHOA) regulating the actin cytoskeleton to affect stress fiber formation and focal complex maturation into focal adhesions. Our results indicate that the disruption of cell migration and impaired normal palatal development by inhibition of *Cxcr4* expression might be mediated through *Rac1* with *RhoA*. The combination of retinoic acid and plerixafor might increase the incidence of cleft palate, which also provided a rationale to guide the use of the drug during conception.

## 1. Introduction

Cleft palate (CP) occurs as a result of abnormalities in the development of the palate [1]. Patients with a cleft palate may not be able to complete the processes of eating, pronunciation, hearing, and so on [2]. As most of CP results from abnormal development of the secondary palate, which is formed by the development of the maxillary process bilaterally, its development is particularly important during the palatogenesis [3]. Firstly, the palatal shelves of the secondary palate grow vertically and are located on the side of the developing tongue [4], then they reposition and elevate above the tongue, growing backwards. Immediately afterward, the palatal shelves grows towards the midline, touching each other and eventually fusing [5]. Disruption at any stage of this complex biological process can result in cleft palate [3]. Due to its intricate etiology, CP has been considered a polygenic disease with both genetic and environmental factors [6]. In recent years, genetic studies on cleft palate have shown that some candidate genes are related to an increased risk of CP, such as msh homeobox 1 (*MSX1*), paired box 9 (*PAX9*) [7], interferon regulatory factor 6 (*IRF6*), transforming growth factor alpha (*TGFA*), bone morphogenetic protein 4 (*BMP4*), fibroblast growth factor 8 (*FGF8*), fibroblast growth factor receptor 2 (*FGFR1*), forkhead box E1 (*FOXE1*) and so on [3,8].

At present, CP is mainly treated using surgery and other methods, such as NAM-plate used in presurgical procedure, which is a device used to stimulate maxillary growth and alter the growth pattern of the patient with CP [9]. However, it still greatly impacts the physical and mental development together with the economic status of patients. Hence, the prevention of cleft palate remains a top priority and the use of drugs during pregnancy is particularly important. There are diverse studies on the etiology of cleft palate using a mice model, and since the main component cells of the palate shelves are the embryonic palatal mesenchymal cells, studies on mouse embryonic palatal mesenchymal (MEPM) cells mainly focus on cell proliferation, apoptosis, migration, osteogenesis and so on, which are important during palatogenesis [10,11]. During embryogenesis, migration plays an important role in development, regeneration, and diseases because the development of tissues and organs depends on the precise migration of progenitor cells from their sites of origin [12,13]. Nevertheless, the regulation on migration, especially MEPM cells in the palate development, is still not clear.

The migratory movement of cells involves multiple aspects, including cell polarity, cytoskeletal reorganization, and signal transduction pathways, all coordinated through interactions between the Rho GTPases [14]. Small guanosine triphosphate-binding proteins make up the Rho family, which has a molecular weight in the range of 20 to 40 kDa [15]. It is known that ras homolog family member A (RHOA), Rac family small GTPase 1 (RAC1), and cell division cycle 42 (CDC42) are the best-characterized members of the Rho family, each belonging to one of three subfamilies [16]. During cell migration, RAC1 mediates cytoskeletal reorganization by controlling lamellipodia formation and focal complex assembly, CDC42 operates cell movement by regulating filopodia formation, and RHOA handles stress fiber formation and focal complex maturation to focal adhesions [17,18].

C-X-C motif chemokine receptor 4 (CXCR4) is a G protein-coupled receptor that participates in cell homing and chemotaxis in hematopoietic and immune systems [19]. CXCR4 is widely expressed in human tissues, such as corneal [20], lung [21],bone marrow [22], etc. It is involved in a variety of biological processes, including angiogenesis, embryonic development, and the regulation of hematopoietic stem cell homing [23], while mice with homozygous knockout of *Cxcr4* will die before birth [24]. Therefore, CXCR4 is a key regulatory gene for normal development. The ligand of CXCR4, stromal cell-derived factor-1 (SDF-1, also known as CXCL12) promotes revascularization and mobilization of hematopoietic cells by binding to CXCR4 on vascular cells, thus accelerating the revascularization of ischemic organs [25]. CXCR4 and its ligand CXCL12, also have been reported to primarily regulate the migration of muscle progenitor cells [13]. However, how *Cxcr4* affected palatal development is unclear. This study aimed to investigate the regulation of *Cxcr4* during palate development and provide a new etiology for the mechanism of cleft palate formation, in addition to offering a scientific basis for the prevention of cleft palate.

## 2. Results

### 2.1. The Migration of the MEPM Cells Was Inhibited by Cxcr4 Knockdown and Was Partially Rescued by the Addition of CXCL12

At key time points in palate development, we extracted palate tissues at E12.5, E13.5, E14.5, E15.5, and E16.5. qRT-PCR was used to detect the expressions of *Cxcr4* and its ligand *Cxcl12*. The results show that there was a high level of *Cxcr4* and *Cxcl12* mRNA expression at E13.5, so we selected MEPM cells from day E13.5 in subsequent experiments (Figure 1A). To determine the role of *Cxcr4* in palatal development, we examined the effect of *Cxcr4* on the proliferation, migration and osteogenic capacity of MEPM cells by knocking down its expression. We first used lentivirus to knock down *Cxcr4* in MEPM cells and verified the knock-down efficiency in MEPM cells by qRT-PCR and Western blot assays (Figure 1B). Subsequently, the proliferation of MEPM cells with *Cxcr4* knockdown was measured by the CCK-8 assay. However, there was no significant difference between the *Cxcr4sh* group and the control group (Figure 1C). We then performed scratch healing and transwell experiments to detect the migration capacity. The results show that the knockdown of *Cxcr4* expression significantly inhibited the migration of MEPM cells, and the inhibited migration effect by the knockdown of *Cxcr4* was partially restored with the addition of CXCL12 protein (Figure 1D–G). Finally, the effect of *Cxcr4* on the osteogenesis of MEPM cells was detected by alizarin red staining and alkaline phosphatase staining assays. The findings show that there was no significant difference between the two groups (Figure 1H,I). These results suggest that the effect of *Cxcr4* on palate development might mainly focus on the regulation of MEPM cell migration.

### 2.2. CXCR4 Inhibitor Plerixafor Blocked the Expression of Cxcr4 and Cxcl12 and Suppressed the Migration of MEPM Cells 

Plerixafor, an inhibitor of CXCR4, was able to effectively interfere with the interaction of CXCR4 with its natural ligand CXCL12. This was confirmed by our results, where qRT-PCR showed that the mRNA expressions of both *Cxcr4* and *Cxcl12* were significantly reduced in the presence of 20 mM plerixafor (Figure 2A). The effect of plerixafor on the proliferation of MEPM cells was also investigated using the CCK-8 assay. Additionally, the plerixafor significantly promoted the proliferation of MEPM cells at 48 and 72 h after the addition of plerixafor (Figure 2B). However, the cell proliferation was reduced at 72 h compared to 48 h. As *Cxcr4* and *Cxcl12* are able to mediate cell migration, we then investigated the effect of plerixafor on the migratory function of MEPM cells using scratch healing and transwell assays. It was found that the migratory function of the cells was significantly inhibited after 48 h (Figure 2C–F), similar to *Cxcr4* knocked down (Figure 1D–G). It confirmed the inhibitory effect of the *Cxcl12*–*Cxcr4* axis on MEPM cell migration byusing plerixafor, CXCR4 inhibitor.

### 2.3. Cxcr4 Knockdown Reduced RAC1, CDC42, and RHOA Expression by Binding to Them, Respectively

As the mechanism of how *Cxcr4* deficiency interfered with MEPM cells’ migratory function was unknown, the effect of *Cxcr4* on the core migration regulators RAC1, CDC42, and RHOA was detected. PCR was used to detect the expressions of *Rac1*, *Cdc42*, and *Rhoa* mRNA, but their expression did not appear to be reduced by the knockdown of *Cxcr4* (Figure 3A). We then speculated that the regulation of RAC1, CDC42, and RHOA using *Cxcr4* might be in the post-transcriptional process and therefore Western blot was used to detect the protein expressions of RAC1, CDC42, and RHOA. It was found that all three proteins were significantly reduced when *Cxcr4* was knocked down and partially recovered with the addition of CXCL12 protein (Figure 3B). Subsequent Co-IP experiments verified that CXCR4, respectively, interacted with RAC1, CDC42, and RHOA, which might regulate their function on MEPM cells through protein combination modulation (Figure 3C).

### 2.4. A Combination of Plerixafor and RA Treatment In Vivo Significantly Increased the Incidence of the Cleft Palate While Reducing the Expression of CXCR4 and Migration-Related Genes

To verify the role of *Cxcr4* in the development of the palate in vivo, its inhibitor plerixafor was used. Pregnant mice were treated with retinoic acid (a common inducer in the CP model) and/or plerixafor, corn oil, and normal saline as a control treatment (Figure 4A). As the number of CP induced by plerixafor alone was not high (6.7%), we then used plerixafor plus 50 mg/kg retinoic acid (RA) to investigate if plerixafor could increase the occurrence of CP, which might directly display the changes of CP in mice and might mimic the drug combination in the clinic. Our results show that after intragastric administration of 50 mg/kg RA to pregnant mice, the incidence of cleft palate in fetal mice was 43.9% (total 6 pregnant mice with 41 fetuses collected, 18 with cleft palate). The incidence of cleft palate in fetal mice increased to 75.6% (total 6 pregnant mice with 41 fetuses collected, 31 with cleft palate) when combined with plerixafor, which was significantly different from the 50 mg/kg RA group alone (Figure 4B). However, when only plerixafor was applied, cleft palate was occasionally observed in fetal mice with a probability of approximately 6.7% (total 6 pregnant mice with 45 fetuses collected, 3 with cleft palate). In addition to taking photographs of cleft palates in fetal mice under a stereomicroscope, we also stained coronal sections of fetal mice heads by H&E staining to observe cleft palate, which was the common method to investigate CP. The H&E-stained sections showed that the bilateral palatal shelves in the CP mice of plerixafor group did not contact and fuse at E16.5 according to normal development; the CP mice of the 50 mg/kg RA group behaved similarly, except that the cleft palate was more severe and the incidence of cleft palate was higher. When 50 mg/kg RA was used in combination with plerixafor, the development of the palatal process was stopped at the elevation stage of the palatal process in CP mice. This indicated that plerixafor could disrupt the normal growth of the palatal process in the early stage of embryonic development, especially combined with RA (Figure 4C). 

To verify whether cleft palate induced by plerixafor in fetal mice had reduced *Cxcr4* gene expression and was associated with impaired migratory function as in vitro, we then performed protein extraction from palatal tissues and examined the expression of CXCR4, additionally with RAC1, CDC42, and RHOA expressions, which are core markers in cell migration in different groups. The results showed that the expression of CXCR4 was the lowest in the group with the group treated with 50 mg/kg RA combined with plerixafor, then only plerixafor and treated with 50 mg/kg RA, and was the highest in the control group, which validated that plerixafor also inhibited CXCR4 expression in vivo during palate development, as in MEPM cells in vitro. Next, CDC42, RAC1, and RHOA were all detected to decrease, similar to CXCR4 expression (Figure 4D). The immunofluorescence staining results also indicate that the expressions of CXCR4, RAC1, and RHOA were reduced in the three treatment groups, respectively (Figure 4E,F). These results in vivo confirm that plerixafor inhibited CXCR4 both in vitro and in vivo in palate shelves and could be used to build a simple CXCR4 low-expression model to investigate CXCR4’s role, which might involve palate development through regulating RAC1 and RHOA expressions.

### 2.5. Knockdown of Cxcr4 Interfered with the Degradation of RAC1, CDC42, and RHOA by Ubiquitin-Proteasome

Next, we aimed to investigate the mechanism for the decrease in protein RAC1, CDC42, and RHOA when *Cxcr4* was knocked down. Autophagy and the ubiquitin-proteasome are the two major systems used to degrade intracellular proteins [26]. Based on previous reports [27], our study focused on whether *Cxcr4* affected the degradation of migration-related genes by the ubiquitin-proteasome system. CHX, an inhibitor of the protein synthesis, was used to detect the half-life of protein. We compared the addition of CHX to the NC group with or without the addition of proteasome inhibition using MG132 to investigate whether the target protein was degraded by the ubiquitin-proteasome system. After the addition of CHX and MG132, it was found that the degradation rates of RAC1, CDC42, and RHOA were significantly lower than only adding CHX. Compared with the NC group, the degradation rate of RAC1 was more rapid in the *Cxcr4sh* group at 3 and 8 h with significant differences; the degradation rate of CDC42 was more rapid in the *Cxcr4sh* group at 6 and 8 h with significant differences; the degradation rate of RHOA was more rapid in the *Cxcr4sh* group at 2 to 8 h with significant differences (Figure 5, Appendix A). This demonstrated that CXCR4 regulated RAC1, CDC42, and RHOA mainly by binding to these proteins and affecting their degradation via the ubiquitin-proteasome system.

## 3. Discussion

Cleft palate (CP) is among the most common birth defects. Patients with cleft palate are often accompanied by lots of symptoms, which seriously affect their life quality and need multidisciplinary care for a long time [28]. As the risk factors of cleft palate mainly include genetic and environmental factors [6], exploring the etiology of CP is important to reduce or intercept its occurrence.

CXCR4 is a conserved protein in the evolutionary process, and the amino acid sequences of humans and mice are highly similar. The expression of CXCR4 in cardiomyocytes, endothelial cells, and smooth muscle cells could be detected on the surface of stem cells [29]. The initial function of CXCR4 discovered is in HIV infection [30] and cancer cell metastasis [31]; its role in development is also very important [29]. During embryogenesis, CXCR4 expressed highly as a chemokine receptor and controls directional migration in many cases. It is detected in the endoderm and is required for the formation of gastrointestinal blood vessels [32]. Second branchial arch skeletal muscle progenitor cells also express *Cxcr4*. The CXCL12/CXCR4 axis has previously been shown to be involved in the development of migrating muscle progenitor cells in the limbs, tongue, pectoral girdle, and cloaca [13]. Research has shown that CXCR4 and immune-related factors might play a role in the development of the normal palate using gene chip analysis and staining with selected antibodies in human embryonic palates [33]. Disruption of cranial neural crest cells migrating to the oral region can lead to cleft lip and palate (CLP), mandibular micrognathia, and glossoptosis in humans [34]. Moreover, the interaction of *Cxcl12*–*Cxcr4* can also promote bone formation and palatal fusion in the development of palate in heme oxygenase (HO)-2 knockout mice [35]. As in our study, there were not significant differences in the cell proliferation and osteogenesis, while a significant migration inhibition was detected when *Cxcr4* was knocked down. We then paid more attention to the role of CXCR4 in the migration of MEPM cells during palatal development. Similar inhibitory effects on cell migration were verified when using the CXCR4 inhibitor plerixafor, and the addition of CXCL12 to MEPM cells with knockdown *Cxcr4* partially restored the inhibited migratory function, suggesting that *Cxcr4* might be involved in palate development by regulating the migration of MEPM cells. 

Previous studies have confirmed that *Rac1* mediates lamellipodia formation and the subsequent assembly of focal complexes [17]; *Cdc42* regulates filopodia formation to control cell movement [18]; *Rhoa* is involved in stress fiber formation and focal complex maturation into focal adhesions at both leading and trailing edges [17]. These three genes are among the most widely studied genes regulating migration and are considered to be the most important genes in the Rho family [36]. In addition, these genes can also directly affect cell migration due to their reduced expressions. Lack of *Cdc42* in telencephalic progenitors causes defects in forebrain development during central nervous system development. *Cdc42* deficiency also caused defects in neuronal polarity and neuron-glia interactions, partly reflected by the fact that *Cdc42*-deficient granule cell precursors had a slowed migration rate through the molecular layer [37]. Cell migration was inhibited by silencing *Rac1*. Conversely, cell motility was restored to normal levels by transfecting *StarD13* (steroidogenic acute regulatory protein-related lipid transfer domain-containing protein 13)—depleted cells with active *Rac1* [38]. VEGF-induced migration is decreased in HUVECs with *Rhoa* knockdown or using negative *Rhoa* mutant [39]. Similar to the results of previous studies, our results show that the protein levels of RAC1, CDC42, and RHOA were all reduced after knocking down *Cxcr4* in MEPM cells and increased after the addition of CXCL12 protein.

To confirm the results in vitro, we set up a simple CXCR4 low-expression model in vivo. Plerixafor (AMD3100), a small molecule CXCR4 antagonist blocking the interaction of local CXCL12 with CXCR4, is the most commonly used CXCL12–CXCR4/CXCR7 targeted agent [40,41]. It was used to inhibit the expression of *Cxcr4* in this study and had a significant inhibitory effect on the migration function of MEPM cells when its concentration was 20 μm. At this concentration, the mRNA expressions of *Cxcr4* and *Cxcl12* were significantly inhibited by plerixafor at 12 and 24 h, but rebounded at 48 h, which was probably due to a negative feedback effect of the cells. We also found a reduced expression of CXCR4 in the palate shelves of the cleft palate in vivo, which suggested that the CXCR4 low-expression CP model by plerixafor was successful. In addition to CXCR4, the expression of RAC1 and RHOA was also significantly reduced in the palate shelf tissue of the cleft palate mice, especially in groups with plerixafor (plerixafor alone and plerixafor plus RA groups), which verified results in vitro and suggested that CXCR4 could regulate the palate development through cell migration by controlling the expressions of RAC1, CDC42, and RHOA. 

As the result of Co-IP suggested the protein of CXCR4 could bind all of three proteins, we wondered how CXCR4 influenced RAC1, CDC42, and RHOA protein expressions. Studies have shown that the degradation of RAC1, CDC42, and RHOA are regulated via several genes through the ubiquitin-proteasome system and their ubiquitination. Atorvastatin inhibited inflammatory cytokine secretion by facilitating proteasome degradation of RAC1 [42], and MG53 could catalyze RAC1 ubiquitination modification [43]; Skp1-Cul1-F-box (SCF) FBXL19 mediated RHOA ubiquitination and proteasomal degradation in lung epithelial cells [44]; X-linked inhibitor of apoptosis protein bound to CDC42 targeting it for proteasomal degradation [45]. We then tested their degradation via the ubiquitin-proteasome pathway by adding MG132 at different times. The results indicate that all three proteins were degraded by the ubiquitin-proteasome system and that their proteasomal degradation was accelerated by knocking down *Cxcr4*, which further confirmed the ubiquitination on RAC1, CDC42, and RHOA protein expression regulation and detected a new target on their modulation. Despite the significant effect of *Cxcr4* knockdown on CDC42 degradation in vitro, CDC42 expression was not changed markedly in the RA group in vivo, which suggested that the regulation of CDC42 was complex and needed further investigation in palate development.

Developmental abnormalities caused by retinoic acid (RA) include short limb deformities (absence of arms and legs, hands and feet directly connected to the trunk), neural tube defects, craniofacial defects including cleft palate, neurocrest cell-derived disorders, and morphological abnormalities of the heart and genitourinary system [46]. For this reason, retinoic acid has been used as a commonly recognized teratogen to study the mechanism of cleft palate. Research has shown that RA induced CP in a dose-dependent manner, with 96.3% of fetal mice having cleft palate when the concentration of RA was 100 mg/kg [47]. Plerixafor has been used in HIV-1 infection, WHIM syndrome, autoimmune diseases and stem cell recruitment [48]. In 2008, it was approved by the US Food and Drug Administration (FDA) and combined with granulocyte colony stimulating factor (G-CSF) as a hematopoietic stem cell mobilization agent for autologous transplantation of patients with non-Hodgkin’s lymphoma (NHL) and multiple myeloma (MM) [49]. During embryonic development, plerixafor treatment reduced the generation rate of retinal organs, damaged the differentiation of ganglion cells and induced morphological changes [50]. Since the use of plerixafor also had an inhibitory effect on palatal development in our preliminary experiment, it was difficult to reveal the teratogenicity of plerixafor on CP occurrence when 100 mg/kg RA was used in combination with plerixafor. So, we reduced the concentration of RA to 50 mg/kg, under which 43.9% of the fetal mice had cleft palate in accordance with the previous studies [51,52]. We then wondered what would happen to CP when 50 mg/kg RA plus plerixafor were used. Interestingly when pregnant mice were treated with plerixafor alone, it could only cause occasional cleft palate (6.7%), while the combination of plerixafor and 50 mg/kg RA can significantly increase the incidence of cleft palate (75.6%), even significantly higher than 50 mg/kg RA alone (43.9%). Although the concentrations of the two drugs are low, the incidence of cleft palate in fetal mice increases significantly owing to their possible synergistic effect. These suggest that even if the dose of drugs used during pregnancy is reduced, the synergistic effects caused by the combination of drugs may increase the risk of abnormal embryonic development.

However, there are still some limitations to this study. Firstly, the conditional knockout mice with *Cxcr4* should be more accurate in investigating the role during palatogenesis. Secondly, autophagy and the ubiquitin-proteasome are the two major systems in degrading intracellular proteins, and we just study the ubiquitin-proteasome on the degradation of RAC1, CDC42, and RHOA after *Cxcr4* knocking down, and whether the mechanism of autophagy plays role in degrading these proteins still needs further study. In addition, the specific sites where CXCR4 regulated the degradation of migratory genes were not clarified in this experiment, which required further investigation.

## 4. Materials and Methods

### 4.1. Cell Culture

MEPM cells were derived from E13.5 ICR pregnant mice as in our previous study [53] (purchased from Sibeifu Company, Beijing, China). The palatal shelves were isolated and then digested with 0.25% trypsin (25200-072, Gibco Thermo Fisher Scientific, Waltham, MA, USA) at 37 °C for 20 min to obtain MEPM cells. Cells were cultured in DMEM/F12 (SV30023.01, HyClone, Logan, UT, USA) containing 10% fetal bovine serum (Gibco Thermo Fisher Scientific, Waltham, MA, USA) and 1% penicillin/streptomycin (C100C5, NCM Biotech, Suzhou. China). The concentration of plerixafor (HY-10046, MCE, Shanghai, China) used in cell culture was 20 μm [54].

### 4.2. Viral Infection

To knock down *Cxcr4*, we subcloned a short hairpin RNA (shRNA) with a sequence complementary to mouse *Cxcr4* into the lentiviral vector with a green fluorescent protein (GFP), and viral packaging was completed by Genechem Company (Shanghai, China). We infected MEPM cells with three types of the above-mentioned viruses along with 6 µg/mL polybrene (Sigma-Aldrich, St. Louis, MO, USA) for 16 h. After 72 h of transfection, MEPM cells were screened with suitable antibiotics [55]. The target sequences for the shRNAs were *Cxcr4* shRNA (*Cxcr4sh*), 5′-GCCTCAAGATCCTTTCCAAAG-3′, and control shRNA (Negative Control), 5′-TTCTCCGAACGTGTCACGT-3′.

### 4.3. Real-Time Reverse Transcriptase Polymerase Chain Reaction (qRT-PCR)

According to the manufacturer’s instructions, total RNA was extracted from the cells using TRIzol reagent (Invitrogen). The TRIzol reagent was added to the samples and incubated for 5 min. After centrifugation (12,000 rpm, 4 °C for 10 min) with chloroform, the lysate was separated into aqueous and organic phases, leaving the RNA in the aqueous phase. Ethanol and a subsequent eluent were used to recover RNA from the aqueous phase. Reverse transcription was performed according to the manufacturer’s protocol using HiScript III All-in-one RT SuperMix Perfect for qPCR (R333-01, vazyme, Nanjing, China). Target genes were amplified in triplicate in real-time using the cDNA and the MagicSYBR Mixture PCR kit (CW3008H, CWBIO, Taizhou, China) in BioRad iCycler iQ real-time PCR detection system or RocGene Archimed-X6 real-time PCR detection system. The conditions were: Hold was 95 °C for 10 min. The cycle was 40 cycles of 95 °C for 30 s, 60 °C for 1 min, and 72 °C for 30 s [56]. The genes were analyzed for expression. The sequences of the primers are listed in Table 1.

### 4.4. Western Blot Analysis

The MEPM cells or tissues of palatal shelves were collected in centrifugal tubes, and the radioimmunoprecipitation assay (RIPA) buffer (C1053, Applygen, Beijing, China) was added for lysis on ice. The total protein was collected after high-speed centrifugation. The protein samples were denatured and electrophoresed on a 10% sodium dodecyl sulphate (SDS) polyacrylamide gel and then transferred using semidry transfer apparatus (BioRad, Hercules, CA, USA) to a polyvinylidene difluoride (PVDF) membrane. The PVDF membrane was blocked in 5% skimmed milk for 1 h, followed by adding primary antibody and overnight incubation [57]. The primary antibodies used were polyclonal anti-CXCR4 (1:1000, Cat. No. 11073-2-AP, Proteintech, Wuhan, China), polyclonal anti-RAC1 (1:1000, Cat. No. 24072-1-AP, Proteintech, Wuhan, China), polyclonal anti-CDC42 (1:1000, Cat. No. 10155-1-AP, Proteintech, Wuhan, China), polyclonal anti-RHOA (1:1000, Cat. No. 10749-1-AP, Proteintech, Wuhan, China) and β-ACTIN antibody (1:2000, Cat. No. 20536-1-AP, Proteintech, Wuhan, China).

### 4.5. Scratch Healing Assays

Cell migration ability was observed using scratch healing assays. MEPM cells were seeded in 6-well plates at a density of 4 × 10^5^ cells per well and DMEM/F12 medium supplemented with 10% FBS. The next day, MEPM cells were scratched vertically and horizontally with the needle tip (three lines per well). Then, DMEM/F12 without FBS was added for subsequent cell culture. After the medium had been changed, cells were cultured for 48 h. Pictures were taken under the microscope. Images were taken at 40× magnification using a microscope at 0 and 48 h. At 40× magnification, 3 random areas in 3 wells of each group were collected and counted using ImagePro Plus 6.0 software [58].

### 4.6. Transwell Assays

After the cells were trypsinized and separated into single cells, the single cell suspension was seeded on transwell chambers (3422, Corning, New York City, NY, USA) with a pore size of 8 μm (2 × 10^4^ cells per well in DMEM/F12). The plate beneath the chambers was prefilled with DMEM/F12 medium containing 15% fetal bovine serum and the chambers were placed in 24-well plates. After 48 h of incubation, the chambers were washed with PBS and fixed in 4% paraformaldehyde solution for 1 h. After another wash with PBS, the chambers were stained with aqueous crystal violet (0.5%) for 10 min and any cells that did not migrate through the polycarbonate (PC) membrane above the chambers were wiped off with a cotton swab. The number of migrating cells was counted at 100× magnification. The microscope was used to count 3 random areas in 3 wells of each group at 100× magnification [56].

### 4.7. Alkaline Phosphatase (ALP) Staining

BCIP/NBT Alkaline Phosphatase Color Development Kit (C3206, Beyotime, Shanghai, China) was used to detect ALP activity. Cells were plated in 12-well plates and cultured in an osteogenic induction medium for 7 days. After the medium was removed, and the cells were fixed with 4% PFA for 30 min. The cells were then incubated overnight at room temperature with 500 μL of working stain [56].

### 4.8. Alizarin Red Staining (ARS)

Osteogenic calcium nodules were detected using the ARS staining kit (C0148S, Beyotime). Cells were plated in 12-well plates and cultured in an osteogenic induction medium for 4 weeks. The medium of plates was then removed and washed with PBS. A total of 1 mL of fixative solution was added and kept at 4 °C for 30 min. The staining solution was added and kept for 1 h. Finally, the images were captured with a light microscope after the fixative solution was removed [59].

### 4.9. Cell Counting Kit-8 (CCK-8) Assay

MEPM cell proliferation was measured by CCK-8 assay (HY-K0301, MCE, Shanghai, China). MEPM cells were plated in 96-well plates. On the next day, Plerixafor was added for 12, 24, and 48 h, after which the medium was removed and the working solution was added to each well according to the manufacturer’s protocol. The multimode detection platform (SpectraMax Paradigm, Molecular Devices, San Jose, CA, USA) was then used to detect absorbance at 450 nm [58].

### 4.10. Animals Handling, Dosing, and Embryo Extracting

All animal experimentation was approved by the Animal Care and Use Committee at Beijing Stomatological Hospital, affiliated with Capital Medical University (permit number: KQYY-202208-003, Beijing, China). C57BL/6J (C57) mice, about 8 weeks of age, were purchased from Sibeifu Company. The control group was given corn oil by gavage and normal saline by intraperitoneal injection; the plerixafor group was given corn oil by gavage and plerixafor 5 mg/kg by intraperitoneal injection; the RA group was given 50 mg/kg of retinoic and normal saline by intraperitoneal injection; the RA and plerixafor combined treatment group (RA+ plerixafor group) was given 50 mg/kg of retinoic and plerixafor 5 mg/kg by intraperitoneal injection.

Pregnant mice at E10.5 were given a concentration of 50 mg/kg of retinoic acid or corn oil by gavage. Intraperitoneal injecting plerixafor or normal saline in pregnant mice were administered once a day for 3 days (E10.5-E12.5) [60].

The control group included 6 pregnant mice, and 45 fetuses were collected; the plerixafor group included 6 pregnant mice, and 45 fetuses were collected; the RA group included 6 pregnant mice, and 41 fetuses were collected; the RA and plerixafor combined treatment group included 6 pregnant mice, and 41 fetuses were collected. All fetuses were used to count the incidence of cleft palate, and 3 fetuses of the control and 3 fetuses with CP when sectioned in other groups, respectively, were used for staining. 

E16.5 stage embryos were dissected in 1× phosphate-buffered saline (PBS). The embryo head was incised, and the upper brain region was removed with forceps. Excess tissue outside the palatal frame was taken away. Finally, the palatal shelves were observed under a stereomicroscope (Lecia S9 D, Wetzlar, Germany), and photographs were taken with a microscope camera (ANDO 2000, Changsha, China). 

The heads of the fetal mice in each group were fixed in a 4% paraformaldehyde solution and dehydrated, embedded, sectioned, and stained [61].

### 4.11. Hematoxylin and Eosin (H&E) Staining

After drying, the sections were dewaxed in xylene and then placed in gradient concentrations of ethanol for hydration. They were then washed with distilled water and stained with haematoxylin for 20 s. After rinsing with water for 5 min, they were placed in differentiation liquid. After 30 s of eosin staining, they were dehydrated, transparentized, and then sealed with neutral resin. Sections were examined microscopically when dried [56]. 

Pictures were observed and captured using OLYMPUS BX61 microscope with Roper Photometrics CoolSnap HQ CCD camera (Japan), and OLYMPUS cellSens Standard was the acquisition software.

### 4.12. Immunofluorescence (IF) Staining

The dewaxed sections were incubated in citrate antigen retrieval solution (Biosharp, Hefei, China) at 95–100 °C for about 20 min. After cooling to room temperature, the sections were washed in PBS and blocked with 5% goat serum for 1 h, then incubated for 12 h at 4 °C with primary antibodies: anti-CXCR4 antibody (1:100), anti-RAC1 antibody (1:100), and anti-RHOA antibody (1:100). Next, sections were incubated with FITC-conjugated secondary antibodies (ZF-0311, Zsbio, Beijing, China) for 1 h at room temperature in the dark. Sections were washed three times in the dark with PBS for 3 min each time. Sections were sealed under coverslips using FluoroshieldTM with DAPI (F6057, Sigma-Aldrich, Zwijndrecht, The Netherlands, USA) seals and analyzed using fluorescence microscopy [57]. Fluorescence intensity was analyzed and quantified by Image J.

### 4.13. Co-Immunoprecipitation, Co-IP

Cells were lysed using Pierce™ IP Lysis Buffer (87788, Gibco Thermo Fisher Scientific, Waltham, MA, USA). The total cell lysate (800 μg protein) was co-incubated with Protein A/G beads, then the indicated antibodies were added and left overnight at 4 °C. After four washes, the immunocomplexes were boiled in SDS sample buffer for 10 min. Co-precipitates were analyzed by SDS-PAGE and subjected to immunoblot analysis [62]. Data were counted and collected by Image J software.

### 4.14. Cycloheximide and MG132 Treatment

For the determination of protein degradation, cells were incubated with the protein synthesis inhibitor cycloheximide (CHX, 10 μg/mL, Sigma-Aldrich) for the indicated times, and the expression of the indicated proteins was evaluated by immunoblotting and quantitative analysis.

To investigate whether the protein was degraded by the ubiquitin-proteasome system (UPS), the cells were treated with CHX (10 μg/mL) plus the proteasome inhibitor MG132 (10 μmol/L, MCE, Shanghai, China) for the indicated times, and the expression of the protein was determined with immunoblotting and quantitative analysis [63]. 

### 4.15. Statistical Analysis

Statistical analysis was performed using GraphPad Prism 8 (GraphPad, San Diego, CA, USA). Student’s *t*-test was used to analyze comparisons between two groups. One-way ANOVA, followed by Tukey’s multiple comparison tests, was used to test the significance of differences between more than two groups. *p* < 0.05 was considered significant.

## 5. Conclusions

In conclusion, we found that *Cxcr4* played roles in palate development mostly by regulating MEPM cell migration. It could bind to the downstream migration-regulating markers RAC1 and RHOA and influence their degradation through the ubiquitin-proteasome system. The combination of retinoic acid (RA) and CXCR4 inhibitor plerixafor might increase the incidence of cleft palate by blocking CXCR4, which also provided a rationale to guide the use of the drug during conception. Further detailed investigation is needed to determine the specific sites at which CXCR4 regulates RAC1 and RHOA degradation via the ubiquitin-proteasome system.

## Figures and Tables

**Figure 1 ijms-24-12740-f001:**
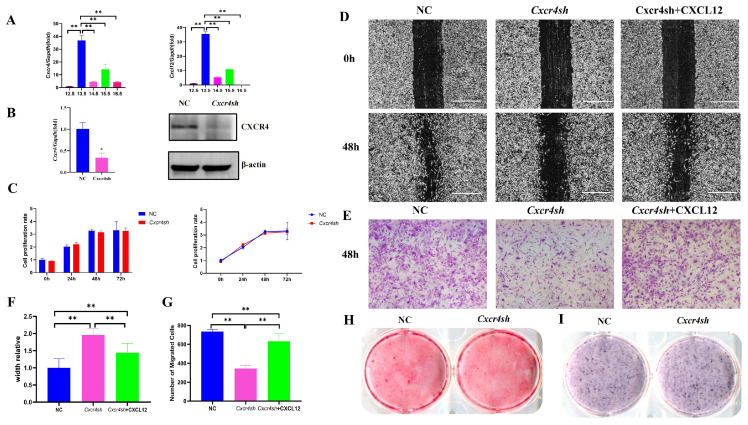
*Cxcr4* regulated migration but not proliferation and osteogenic capacity of MEPM cells. (**A**) *Cxcr4* and *Cxcl12* were expressed at E12.5, E13.5, E14.5, E15.5 and E16.5 but were highest at E13.5. (**B**) The knock-down efficiency of *Cxcr4* in MEPM cells by qRT-PCR and Western blot assays. Mean ± SD, * *p* < 0.05. (**C**) CCK-8 was used to detect the difference between NC and *Cxcr4sh* groups. ns: not significant. Migration capacity of MEPM cells, which was divided into three groups, including the NC group, *Cxcr4sh* group, and the *Cxcr4sh* + CXCL12 group, was measured via scratch healing assay (**D**) and transwell assay at 48 h (**E**). Scale bar: 500 µm. (**F**,**G**) Quantifications of scratch healing assay (**F**) and transwell assay (**G**). ** *p* < 0.01. The data shown here are from representative experiments with 3 biological replicates and 3 technical replicates. (**H**,**I**) The osteogenic capacity of MEPM cells was detected by alizarin red staining (**H**) and alkaline phosphatase staining assay (**I**).

**Figure 2 ijms-24-12740-f002:**
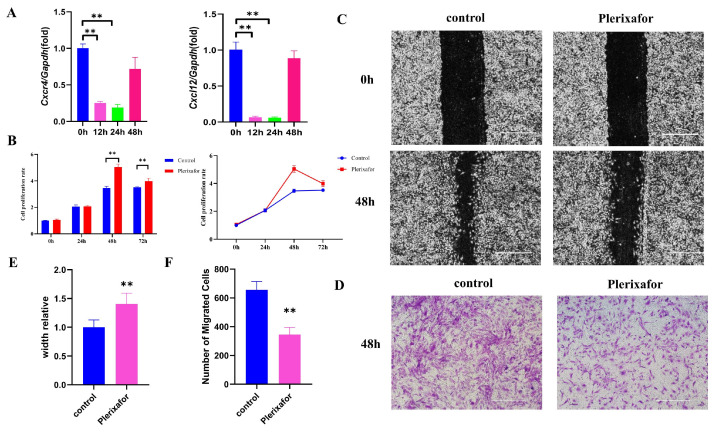
Plerixafor inhibited the expressions of *Cxcr4* and *Cxcl12* and suppressed the migration of MEPM cells. (**A**) Plerixafor (20 μm) treatment significantly inhibited the mRNA expression of *Cxcr4* and *Cxcl12* at 12 and 24 h, which was measured by qRT-PCR. (**B**) CCK-8 was used to detect the influence of MEPM cell proliferation after plerixafor (20 μm) treatment at 12, 24, and 48 h. ** *p* < 0.01. Migration capacity of MEPM cells after plerixafor (20 μm) treatment was measured via scratch healing assay (**C**) and transwell assay at 48 h (**D**). Scale bar: 500 µm. (**E**,**F**) Quantifications of scratch healing assay (**E**) and transwell assay (**F**). ** *p* < 0.01. The data shown here are from representative experiments with 3 biological replicates and 3 technical replicates.

**Figure 3 ijms-24-12740-f003:**
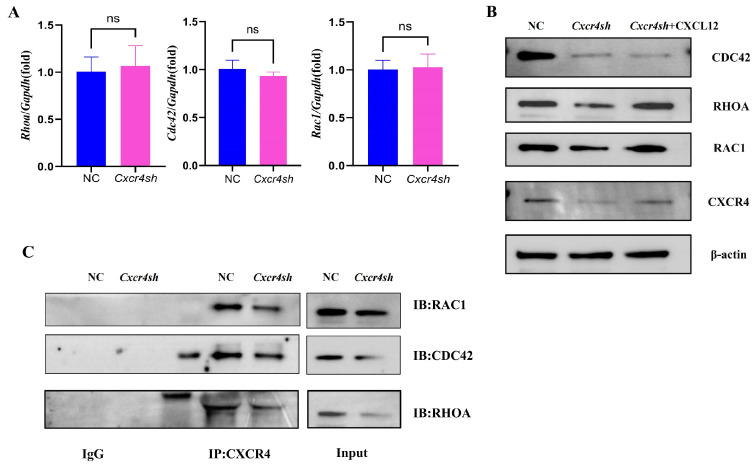
Knockdown of *Cxcr4* downregulated the expressions and bindings of the proteins of migration-regulating markers. (**A**) The mRNA levels of *Rac1*, *Cdc42*, and *Rhoa* in the NC group compared to the *Cxcr4sh* group. ns: not significant. (**B**) The protein levels of RAC1, CDC42, and RHOA compared to the NC, *Cxcr4sh*, and *Cxcr4sh* + CXCL12 groups. (**C**) Anti-CXCR4 coprecipitates from MEPM cells were, respectively, analyzed with anti-RAC1, anti-CDC42, and anti-RHOA antibodies to verify the interaction between CXCR4 and those three proteins regulating cell migration.

**Figure 4 ijms-24-12740-f004:**
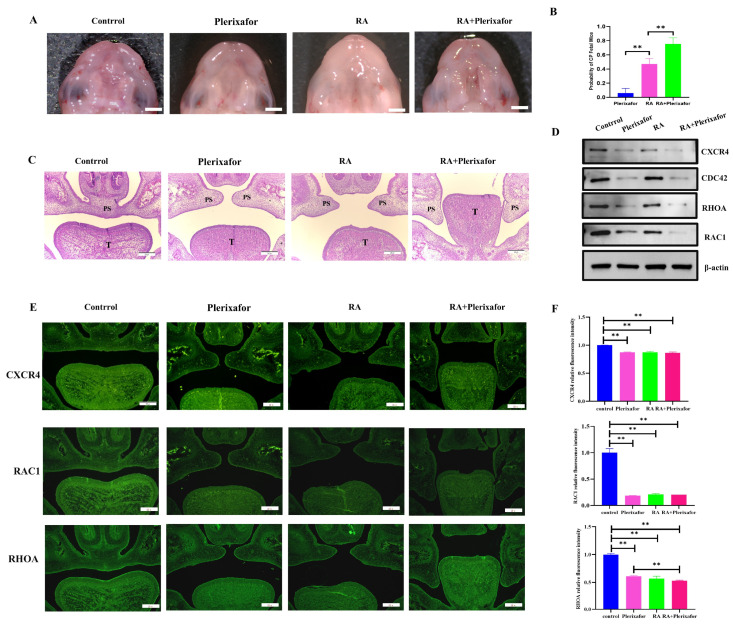
Combined treatment with plerixafor increased the incidence of cleft palate caused by RA only and suppressed the expressions of CXCR4 and proteins involved in migration in vivo. (**A**) Fetal mice with cleft palate were observed by stereomicroscope. Scale bar: 1 mm. (**B**) GraphPad Prism software was used to analyze the incidence of cleft palate in fetal mice. (**C**) The H&E staining of palate shelves at E16.5. Scale bar: 200 μm. PS represents palatal shelves and T represents togue. ** *p* < 0.01. (**D**) Western blot was used to measure the protein expression of CXCR4, RAC1, CDC42, and RHOA in different groups of palatal tissues. (**E**) The expression of CXCR4, RAC1, and RHOA located in the palate was detected using immunofluorescence staining, and positive expression showed green fluorescence. (**F**) Statistical analysis of fluorescence intensity. Scale bar: 200 μm. ** *p* < 0.01.

**Figure 5 ijms-24-12740-f005:**
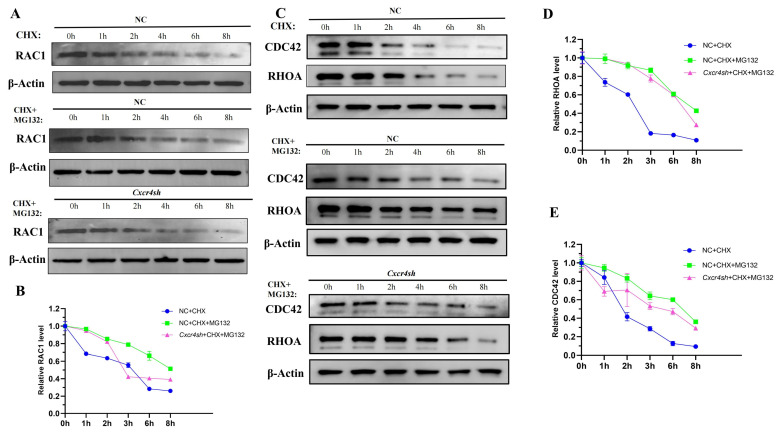
CXCR4 regulates the protein degradation of RAC1, CDC42, and RHOA through the ubiquitin-proteasome pathway. (**A**,**B**) Quantitative analyses of RAC1 protein stability in the NC group in the presence of the protein synthesis inhibitor cycloheximide (CHX) (10 μg/mL) for the indicated times are shown (*n* = 3). Quantitative analyses of RAC1 degradation in the NC group and *Cxcr4sh* in the presence of CHX and MG132 are shown (*n* = 3). β-actin was used as a loading control for Western blot. The proteins were quantified by Image J; (**C**–**E**) Quantitative analyses of CDC42 and RHOA protein stability in the NC group and *Cxcr4sh* group in the presence of CHX for the indicated times are shown (*n* = 3). Quantitative analyses of CDC42 and RHOA degradation in the presence of CHX and MG132 are shown (*n* = 3). β-actin was used as a loading control for Western blot. The proteins were quantified by Image J.

**Table 1 ijms-24-12740-t001:** The primer sequences used for quantitative real-time RT-PCR.

Gene	Primer Sequence (5′-3′)
*Cxcr4*-F	GAAGTGGTCTGGAGACTAT
*Cxcr4*-R	TTGCCGACTATGCCAGTCAAG
*Cxcl12*-F	TGCATCAGTGACGGTAAACCA
*Cxcl12*-R	TTCTTCAGCCGTGCAACAATC
*Rac1*-F	GAGACGGAGCTGTTGGTAAAA
*Rac1*-R	ATAGGCCCAGATTCACTGGTT
*Cdc42*-F	ATTATGACAGACTACGACCGCT
*Cdc42*-R	AGTGGTGAGTTATCTCAGGCA
*Rhoa*-F	CTCTCTTATCCAGACACCGATGT
*Rhoa*-R	TGTGCTCGTCATTCCGAAGG
*Gapdh*-F	AGGTCGGTGTGAACGGATTTG
*Gapdh*-R	TGTAGACCATGTAGTTGAGGTCA

## Data Availability

All datasets generated for this study are included in the article/Appendix A; further inquiries can be directed to the corresponding author.

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
