# Peer review of "Inhibition of Cxcr4 Disrupts Mouse Embryonic Palatal Mesenchymal Cell Migration and Induces Cleft Palate Occurrence"

_ijms, 2023, doi:10.3390/ijms241612740_

Round 1
Reviewer 1 Report
While this study points to some interesting novel findings, I would recommend the following revisions prior to further consideration for publication: The authors’ Introduction section (particularly, the first paragraph) lacks appropriate citation of prior work supporting each of the statements they make. For example, the first reference is not made to any published literature until line 38. Each statement should be supported by verified literature, which is readily available on all these topics discussion in the Introduction. I would also encourage the authors to rigorously read the prior literature they are referencing to better fine-tune their introductory statements, as many of them are lukewarm without much detail or meaning. This would greatly enhance the framework of the manuscript for potential readers.
Next, the authors do not include a description of justification for the dosages of the cleft-inducing therapeutics used in their methodology. While the use of RA is quite standard in the field to induce a cleft phenotype, I would hope the authors followed conventional guidelines for this technique from prior literature. However, for plerixafor, I would like to know if the authors delivered an equivalent dose in the pregnant dams to the level of bioactive therapeutic is delivered to humans who take this medication routinely. Furthermore, based on the Methods section, it appears the authors only treated each pregnant dam once per gestation with each respective treatment. Is this single therapeutic loading sufficient and/or translatable to the level of therapeutic loading clinically performed on human patients?
The palatal shelves are composed of several different tissue types, stemming from both epithelial and mesenchymal cell lineages. As the authors did not perform any steps in their protocol to separate the epithelium from the mesenchyme of the palatal shelves analyzed - neither in the ex vivo primary culture systems nor the in vivo post-treatment histomorphometry - it is difficult to make a claim as the to cell population most affected by the knockdown of Cxcr4. The authors should address this shortcoming and modify their conclusions accordingly.
The Methods are well described and appear to be sound scientifically. Again, the authors should temper their statements in their Results and Discussion based on the data that they present.
Reviewer 2 Report
Dear Authors,
Good experimental work indeed. I have only some small comments:
1) Materials and methods. Please, indicate how many animals were involved in the subsection 4.10. How many were in each experimental group,, in control group, how many were pregnant mice and how many embryons were obtained, and how many were used from the obtained ones.
2) give, please, the references for the all methods described, especially in subsections 4.111, 4.12 and 4.13.
3) please, add the Limitation paragraph at the end of Discussion.
4) one reference (out of 37) is from the previous century. Are you sure that you really need it and it cant be replaced by more modern one?
Author Response
Thank you very much for reviewing our manuscript “Inhibition of Cxcr4 disrupts mouse embryonic palatal mesenchymal cell migration and induces cleft palate occurrence”. We are very grateful to you your helpful feedback on our work. In response to your suggestions, we have revised the article and indicated these changes in the manuscript. In addition, the following are responses to your comments.
Good experimental work indeed. I have only some small comments:
Point 1: Materials and methods. Please, indicate how many animals were involved in the subsection 4.10. How many were in each experimental group, in control group, how many were pregnant mice and how many embryons were obtained, and how many were used from the obtained ones.
Response 1: Thank you sincerely for your valuable suggestions. The control group included 6 pregnant mice, and 45 fetuses were collected; the plerixafor group included 6 pregnant mice, and 45 fetuses were collected; the RA group included 6 pregnant mice, and 41 fetuses were collected; The RA and plerixafor combined treatment group included 6 pregnant mice, and 41 fetuses were collected. All fetuses were used to count the incidence of cleft palate, and 3 fetuses of the control and 3 fetuses with CP when sectioned in other groups respectively were used for staining. We have modified it in Materials and methods. 4.10. of the article which labelled in yellow color.
Point 2: give, please, the references for the all methods described, especially in subsections 4.111, 4.12 and 4.13.
Response: Thank you very much for pointing out this. We have added the references for the all methods described in the article.
Point 3: please, add the Limitation paragraph at the end of Discussion.
Response 3: Thank you very much for the suggestions. We have added the limitation paragraph at the end of Discussion.
“However, there are still some limitations to this study. Firstly, the conditional knockout mice with Cxcr4 should be more accurate in investigating its role during pala-togenesis. Secondly, autophagy and the ubiquitin-proteasome are the two major systems used to degrade intracellular proteins and we just study the ubiquitin-proteasome on degradation of RAC1, CDC42, and RHOA after Cxcr4 knocking down, the mechanism of autophagy still needs further study. In addition, the specific sites where CXCR4 regulated the degradation of migratory genes were not clarified in this experiment, which required further investigation.” (Stated on page 10, lines 353-360)
Point 4: one reference (out of 37) is from the previous century. Are you sure that you really need it and it cant be replaced by more modern one?
Response 4: Thank you very much for your comments. The corresponding sentence in the paragraph has been revised as “The function of CXCR4 discovered is in initial step of HIV infection [30] and cancer cell metastasis [31].” stated on page 8, lines 256-257.
Reference
- Masenga, S. K.; Mweene, B. C., et al., HIV-Host Cell Interactions. Cells 2023, 12 (10). PMID: 37408185 DOI: 10.3390/cells12101351
- Marayati, R.; Julson, J., et al., PIM3 kinase promotes tumor metastasis in hepatoblastoma by upregulating cell surface expression of chemokine receptor cxcr4. Clin Exp Metastasis 2022, 39 (6), 899-912. PMID: 36315303 DOI: 10.1007/s10585-022-10186-3
Reviewer 3 Report
Dear Authors, Thank you for this interesting Article to review. In generał, I rate It high, but here are some suggestions:
1. In line 41, Please szpeciły which methods, eg. NAM plate, see
Paradowska-Stolarz A, Mikulewicz M, Duś-Ilnicka I. Current Concepts and Challenges in the Treatment of Cleft Lip and Palate Patients-A Comprehensive Review. J Pers Med. 2022 Dec 19;12(12):2089. doi: 10.3390/jpm12122089. PMID: 36556309; PMCID: PMC9783897.
2. Please, write in the introductiin the paragraph on complex genetics od clefts, in particular MSX1, PAX9, , IRF6, TGNF ~ those arę the basics od clefts genetics etiology. Please Find more than onły: Nasroen SL, Maskoen AM, Soedjana H, Hilmanto D, Gani BA. IRF6 rs2235371 as a risk factor for non-syndromic cleft palate only among the Deutero-Malay race in Indonesia and its effect on the IRF6 mRNA expression level. Dent Med Probl. 2022;59(1):59–65. doi:10.17219/dmp/142760
3. In M&M, Please add the permissions numbers for this kind of research
4. Add the limitations And try to specify more than 1 conclusion
5. Linę 229, verify the abbreviations. I would resign here from speculations on co-existing syndromes (or sequences), because many of them arę genetically driven and this sentence may bend the truth or cause misconvenience
thank you
Author Response
Thank you very much for reviewing our manuscript “Inhibition of Cxcr4 disrupts mouse embryonic palatal mesenchymal cell migration and induces cleft palate occurrence”. We are very grateful to you your helpful feedback on our work. In response to your suggestions, we have revised the article and indicated these changes in the manuscript. In addition, the following are responses to your comments.
Dear Authors, Thank you for this interesting Article to review. In generał, I rate It high, but here are some suggestions:
Point 1: In line 41, Please szpeciły which methods, eg. NAM plate, see
Paradowska-Stolarz A, Mikulewicz M, Duś-Ilnicka I. Current Concepts and Challenges in the Treatment of Cleft Lip and Palate Patients-A Comprehensive Review. J Pers Med. 2022 Dec 19;12(12):2089. doi: 10.3390/jpm12122089. PMID: 36556309; PMCID: PMC9783897.
Response 1: Thank you very much for your valuable suggestions. We have revised them as follows which labelled with yellow color: At present, CP is mainly treated by surgery and other methods, such as NAM-plate used in presurgical procedure which is a device used to stimulate maxillary growth and alter the growth pattern of the patient with CP [9].
Reference
[9]. Paradowska-Stolarz, A.; Mikulewicz, M., et al., Current Concepts and Challenges in the Treatment of Cleft Lip and Palate Patients-A Comprehensive Review. J Pers Med 2022, 12 (12).
Point 2: Please, write in the introduction the paragraph on complex genetics od clefts, in particular MSX1, PAX9, IRF6, TGNF ~ those arę the basics od clefts genetics etiology. Please Find more than onły: Nasroen SL, Maskoen AM, Soedjana H, Hilmanto D, Gani BA. IRF6 rs2235371 as a risk factor for non-syndromic cleft palate only among the Deutero-Malay race in Indonesia and its effect on the IRF6 mRNA expression level. Dent Med Probl. 2022;59(1):59–65. doi:10.17219/dmp/142760
Response 2: Thank you sincerely for your valuable suggestions. We have revised the article in line 40 to 45 as follows which labelled with yellow color: In recent years, genetic studies on cleft palate have shown that some candidate genes that are related to an increased risk of CP, such as msh homeobox 1 (MSX1), paired box 9 (PAX9) [7], interferon regulatory factor 6 (IRF6), transforming growth factor alpha (TGFA), bone morphogenetic protein 4 (BMP4), fibroblast growth factor 8 (Fgf8), fibroblast growth factor receptor 2 (FGFR1), forkhead box E1 (FOXE1) and so on [3, 8].
Reference
[3] Bush, J. O.; Jiang, R., Palatogenesis: morphogenetic and molecular mechanisms of secondary palate development. Development 2012, 139 (2), 231-43. PMID: 22186724 PMCID: PMC3243091 DOI: 10.1242/dev.067082
[7] Jia, S.; Zhou, J., et al., Small-molecule Wnt agonists correct cleft palates in Pax9 mutant mice in utero. Development 2017, 144 (20), 3819-3828. PMID: 28893947 PMCID: PMC5675451 DOI: 10.1242/dev.157750
[8] Nasroen, S. L.; Maskoen, A. M., et al., IRF6 rs2235371 as a risk factor for non-syndromic cleft palate only among the Deutero-Malay race in Indonesia and its effect on the IRF6 mRNA expression level. Dent Med Probl 2022, 59 (1), 59-65. PMID: 35226971 DOI: 10.17219/dmp/142760
Point 3: In M&M, Please add the permissions numbers for this kind of research
Response 3: Thank you very much for your comments. We have provided it in Materials and methods. 4.10. of the revised manuscript as follows which labelled with yellow color (stated on page 12, lines 449-451): All animal experimentation was approved by the Animal Care and Use Committee at Beijing Stomatological Hospital, affiliated with Capital Medical University (permit number: KQYY-202208-003, Beijing, China).
Point 4: Add the limitations And try to specify more than 1 conclusion
Response 4: Thank you sincerely for your valuable suggestions. We have added the limitation paragraph at the end of Discussion. “However, there are still some limitations to this study. Firstly, the conditional knockout mice with Cxcr4 should be more accurate in investigating its role during palatogenesis. Secondly, autophagy and the ubiquitin-proteasome are the two major systems used to degrade intracellular proteins and we just study the ubiquitin-proteasome on degradation of RAC1, CDC42, and RHOA after Cxcr4 knocking down, the mechanism of autophagy still needs further study. In addition, the specific sites where CXCR4 regulated the degradation of migratory genes were not clarified in this experiment, which required further investigation.” (Stated on page 10, lines 353-360)
The Conclusion section has been revised. “In conclusion, we found that Cxcr4 played roles on palate development mostly by regulating MEPM cells migration. It could bind to the downstream migration-regulating markers RAC1 and RHOA and influence their degradation through ubiquitin-proteasome system. The combination of retinoic acid (RA) and CXCR4 inhibitor plerixafor might in-crease the incidence of cleft palate by blocking CXCR4, which also provided a rationale to guide the use of the drug during conception. Further detailed investigation is needed to determine the specific sites at which CXCR4 regulates RAC1 and RHOA degradation via the ubiquitin-proteasome system.” (Stated on page 13, lines 517-524)
Point 5: Linę 229, verify the abbreviations. I would resign here from speculations on co-existing syndromes (or sequences), because many of them arę genetically driven and this sentence may bend the truth or cause misconvenience
Response 5: Thank you very much for the valuable suggestions. To avoid confusion, we have revised the sentences as the following in line 248: Cleft palate (CP) are among the most common birth defects. Patients with cleft palate are often accompanied by lots of symptom which are seriously affect their life quality and need multidisciplinary care for a long time [28]. As the risk factors of cleft palate mainly include genetic and environmental factors [6], exploring the etiology of CP is important to reduce or interceptive its occurrence.
Reference
[28] Shkoukani, M. A.; Lawrence, L. A., et al., Cleft palate: a clinical review. Birth Defects Res C Embryo Today 2014, 102 (4), 333-42. PMID: 25504820 DOI: 10.1002/bdrc.21083
[6] Mossey, P. A.; Little, J., et al., Cleft lip and palate. Lancet 2009, 374 (9703), 1773-85. PMID: 19747722 DOI: 10.1016/S0140-6736(09)60695-4
Round 2
Reviewer 3 Report
Thank you, the article can be accepted in this form